# Efficacy and Safety of Monacolin K Combined with Coenzyme Q10, Grape Seed, and Olive Leaf Extracts in Improving Lipid Profile of Patients with Mild-to-Moderate Hypercholesterolemia: A Self-Control Study

**Nicholas Angelopoulos** [1,2,*] , **Rodis D. Paparodis** [3,4] , **Ioannis Androulakis** [1,5] , **Anastasios Boniakos** [1] , **Panagiotis Anagnostis** [6] , **Vasilis Tsimihodimos** [7] and **Sarantis Livadas** [1]

1   Endocrine Unit, Athens Medical Centre, 65403 Athens, Greece
2   Private Practice, Venizelou Str., 65302 Kavala, Greece
3   Center for Diabetes and Endocrine Research, University of Toledo College of Medicine and Life Sciences, Toledo, OH 43614, USA
4   Private Practice, Gerokostopoulou 24, 26221 Patra, Greece
5   Private Practice, Tzanaki Emmanouil 17, 73134 Chania, Greece
6   Unit of Reproductive Endocrinology, 1st Department of Obstetrics and Gynecology, Medical School, Aristotle University of Thessaloniki, 54124 Thessaloniki, Greece
7   Department of Internal Medicine, Medical School, University of Ioannina, 45110 Ioannina, Greece
*   Correspondence: drangelnick@gmail.com; Tel.: +30-2510225200; Fax: +30-2510830900

**Abstract:** The objective of the present study was to assess the lipid-lowering efficacy and safety of a novel dietary supplement containing monacolin K combined with the coenzyme Q10 and grape seed and olive tree leaf extracts (Arichol®®) on the lipid profile of adults with moderate cholesterol elevations and an absence of concomitant risk factors. We recruited patients from our Endocrinology Clinics in Greece who had low-density lipoprotein cholesterol (LDL-C) 140–180 mg/dL, were on no medications affecting serum lipid concentrations, and consented to participate in the present study. All subjects received 8-weeks supplementation with Arichol®® once daily. We measured total cholesterol (TC), high-density lipoprotein cholesterol (HDL), LDL-C, triglycerides (TG), and liver enzymes with enzymatic colorimetric assays at baseline and at the end of the study, and documented complaints potentially attributable to muscle injury. We recruited a total of 37 subjects, 33 females and 4 males (with a mean age of 55.89 ± 1.50 [mean ± standard error mean, SEM]). The treatment resulted in a statistically significant reduction in TC (from 258.9 ± 4.0 mg/dL to 212.7 ± 4.5 mg/dL, $p < 0.001$), LDL-C (from 173.8 ± 3.5 to 129.0 ± 4.5 mg/dL, $p < 0.001$), and TG (from 127.0 ± 12.2 to 117.0 ± 9.2, mg/dL, $p = 0.012$) concentrations, while HDL-C concentrations remained unchanged. There were no alterations in liver enzymes or symptoms of muscle pain in any subject. These promising results suggest that supplementation with this nutraceutical mixture favorably influences lipid concentrations during a short period of administration while exhibiting an excellent safety profile. Larger controlled studies are required to assess the potential for cardiovascular risk reduction with the above compound.

**Keywords:** monacolin K; coenzyme Q10; grape seed extract; olive leaf extract; hypercholesterolemia

## 1. Introduction

Extracts of red yeast rice (RYR), of which the chief bioactive compound, namely monacolin K, is a weak reversible inhibitor of 3-hydroxy-3-methyl-glutaryl-coenzyme A reductase [1], is currently considered to be the most efficacious cholesterol-lowering nutraceutical. Its safety and efficacy among, in particular, patients suffering from hypercholesterolemia has been demonstrated in a number of large meta-analyses of randomized trials [2–4]. Of note, studies have recently shown that when administered in association

with other bioactive compounds with different mechanisms of action, RYR increases the cholesterol-lowering effect [5,6]. Meanwhile, it has also been reported that the coenzyme Q10 (CoQ10)—a cofactor in the Krebs cycle, which, when absorbed at an adequate amount, is essential for oxidative reduction processes—has the capacity to prevent low-density cholesterol LDL-C peroxidation. Throughout Mediterranean countries, leaves of the olive tree (*Olea europaea* L.) have traditionally been used in herbal remedies [7]. Olive leaf extract (OLE) contains large amounts of phenolic antioxidant, or oleuropein; far greater, in fact, than what is contained in the olive fruit or olive oil [8,9], while it also possesses lipid-lowering properties [10,11]. While the exact mechanism of action of OLE's health-related beneficial effects is not well understood, animal studies have shown that OLE decreases the activity of hydroxymethylglutaryl-CoA reductase, leading to a reduction in cholesterol synthesis in rat hepatocytes [11].

Studies reporting promising results of the antioxidant, lipid-lowering, and anti-atherogenic properties of polyphenols and flavonoids present in GSE have attracted great interest [12]. GSE also contains a high amount of unsaturated fatty acids such as linoleic acid, linolenic acid, oleic acid, and palmitic acid, which are important in lipid metabolism due to their ability to lower both total cholesterol (TC) and LDL-C [13].

High plasma homocysteine, which is an indicator of cardiovascular disease [14], is linked to B vitamin deficiencies, including folate, vitamin B6, and vitamin B12. Meanwhile, although some observational studies have indicated that consumption of foods fortified with folic acid and dietary intake of vitamins B6 and B12 may lead to a reduced risk of cardiovascular disease [15,16], clinical trials studying the possible benefits of vitamin B complex supplementation in cardiovascular disease demonstrated no clear benefits [17–19].

The aim of this study was to assess the lipid-lowering activity and safety of a novel, commercially available dietary supplement (Arichol®®, Epsilon Health) combining PYR, CoQ10, OLE, GSE, and vitamin B in volunteers with moderate elevation of C-LDL levels.

## 2. Materials and Methods

### 2.1. Study Population

Our study is an investigator-initiated protocol whose recruitment base comprised patients at four private Endocrinology Clinics and three tertiary care Endocrinology Clinics all located in Greece. Potential recruits all consisted of consenting patients attending our clinics for different endocrine disorders who were screened for eligibility at the first visit to the clinic. Patients were eligible to participate in the present study if they were adults (>18 years of age), had a fasting LDL-C concentration between 140 and 180 mg/dL despite previous nutritional efforts to lower cholesterol, and a 10-year atherosclerotic cardiovascular disease risk (ASCVD Risk) < 7.5%. The 10-year risk was calculated in all participants (aged >40 years) using the updated ASCVD Risk Estimator Plus (found in: https://tools.acc.org/ASCVD-Risk-Estimator-Plus/#!/calculate/estimate/ (accessed on 10 January 2022)) based on the 2019 ACC/AHA Guideline on the Primary Prevention of Cardiovascular Disease [20], the 2018 ACC/AHA et al. Guideline on the Management of Blood Cholesterol [21], the 2013 ACC/AHA Guideline on the Assessment of Cardiovascular Risk [22], the 2017 ACC/AHA et al. Guideline for the Prevention, Detection, Evaluation, and Management of High Blood Pressure in Adults [23], and the 2016 Million Hearts Longitudinal ASCVD Risk Assessment Tool user guide [24].

Excluded from the study were the following: patients (1) who had received any dose of lipid-altering medications, such as statins, within 8 weeks prior to study enrollment; (2) with an ASCVD risk $\geq$ 7.5%, including all patients with any form of diabetes and those with any degree of known atherosclerotic vascular disease, if they could tolerate any statin dose; (3) with abnormal liver function tests (elevated serum transaminases); with abnormal renal function (Modification of Diet in Renal Disease calculated glomerular filtration rate, MDRD GFR < 60 mL/min); (4) with allergies to any food, including rice, and those with excessive alcohol intake; and (5) pregnant and lactating women and those in their reproductive years not on oral contraceptives.

All study participants received education on the lifestyle modification program (basic prevention and treatment, knowledge of dyslipidemia, smoking cessation, reduction of alcohol intake, over three times per week aerobic exercise) that was to be applied, while they were prescribed a standardized Mediterranean diet characterized by the high consumption of fish, fruit, vegetables, legumes, olive oil, and unrefined whole grains accompanied by a modest intake of lean meats and alcohol. Patients were encouraged to follow the prescribed diet during the study period and compliance was checked via a questionnaire on the final visit. During the study period, blood samples were taken twice; at the beginning of the study after a run-in period of 2 months while being on a diet (baseline visit, $t_0$) and at the end of the study (follow-up visit, $t_8$, 8 weeks after the initiation of the supplement Arichol$^{\text{®®}}$ tablets, Epsilon Health, one tablet every evening after eating; ingredients are shown in Table 1).

**Table 1.** Chemical composition of the RYR preparation daily dose.

| RYR monacolin K | 10 mg |
|---|---|
| CoQ10 | 2 mg |
| Vitamin B5 | 6 mg |
| Vitamin B6 | 1.4 mg |
| Vitamin B2 | 1.4 mg |
| Vitamin B1 | 1.1 mg |
| Folic acid | 200 µg |
| Biotin | 50 µg |
| Vitamin B12 | 2 µg |
| OLE | 50 mg |
| GSE | 50 mg |

*2.2. Measurements*

Our study protocol consisted of 3 days of high-carbohydrate diet meals (>300 g daily), followed by a 10-h overnight fast [25]. On the following morning, all patients were weighed on a standard scale and their height was measured by a qualified physician. Subsequently, venous blood samples, consisting of 25 mL of blood, were collected by a study physician. The serum was separated by centrifugation (1058 RCF, for 15 min at +4 °C, Rotofix 32A centrifuge, Andreas Hettich, GmbH & Co. KG, Tuttlingen, Germany) and stored at −80 °C in a refrigerator for further analysis, which was conducted simultaneously by an external laboratory. Serum lipids (TC, TG, LDL-C, and HDL-C) were measured with an enzymatic colorimetric assay (Dimension Vista$^{\text{®®}}$ 500 System, Siemens, Munich, Germany). Liver enzymes consisting of alanine aminotransferase (ALT), aspartate aminotransferase (AST), gamma-glutamyl transferase (γ-gt), and creatine kinase (CK) were measured using an enzymatic method (, Dimension Vista$^{\text{®®}}$ 500 System Siemens, Munich, Germany). In order to participate in our intervention protocol, all subjects were instructed to take one tablet of Arichol$^{\text{®®}}$ (Epsilon Health, Athens, Greece) daily, ingested after the meal, which consisted of RYR, CoQ10, GSE, OLE, and vitamin B complex for 8 weeks. After this period of time, the procedures that were performed at the baseline visit were repeated. Body mass index (BMI kg/m$^2$) (before and after the intervention) was calculated as the ratio of weight (in kilograms) to squared height (in meters).The study was conducted in accordance with the Declaration of Helsinki and was approved by the Institutional Review Board of the Athens Medical Center General Hospital, Athens, Greece. A safety monitoring board was available to review the data of the study until completion.

*2.3. Statistical Analysis*

Normality of the distribution of our data was analyzed using the Kolmogorov–Smirnov test. A paired t test was used to compare within-group differences from $t_0$ to $t_8$. Differences of data with skewed distributions (TG) were analyzed with the Wilcoxon non-parametric test. Results were presented as mean values ± standard error mean (SEM). Moreover, the difference between time points was indicated in percentage. Statistical Package of Social Sciences 21.0 (SPSS Inc., Chicago, IL, USA) was used for the statistical analysis and *p* values < 0.05 were deemed significant.

## 3. Results

We initially enrolled 43 subjects who completed the baseline visit testing; however, one was removed from the study protocol a few days after initiation of the intervention because of complaints of palpitations, but no specific cardiac diagnosis was established. He is currently in good health. An additional five subjects were lost to follow-up. Four people refused to return at the end of the trial and one person was disengaged because of relocation. Finally, 37 subjects completed the study protocol, consisting of 33 females and 4 males. Prior to study enrollment, 5 of our subjects reported significant complaints of myalgia when introduced to simvastatin 10 mg daily, while the remaining 32 subjects were treatment-naive with regard to any lipid-lowering intervention.

The mean age was 55.9 ± 1.5 years (range 32–72 years). In two patients who were younger than 40 years old (a 36-year-old male and a 32-year-old female), the 10-year ASCVD Risk was not applicable. Demographic and biochemical characteristics are shown in Table 2.

**Table 2.** Baseline characteristics and lipid profile of the study population.

| Variables | Patients (N = 37) |
|---|---|
| Age (years) | 55.9 ± 1.5 |
| Sex (M/F, %) | 11/89 |
| BMI (Kg/m$^2$) | 25.3 ± 0.54 |
| TC (mg/dL) | 258.9 ± 0.4 |
| LDL-C (mg/dL) | 173.8 ± 3.45 |
| HDL-C (mg/dL) | 63.1 ± 2.6 |
| TG (mg/dL) | 127 ± 12.19 |
| AST (U/L) | 27.43 ± 0.99 |
| ALT (U/L) | 30.43 ± 0.99 |
| γ-gt (U/L) | 29.18 ± 1.09 |
| CPK (IU/L) | 75.89 ± 3.22 |

TC: total cholesterol; LDL-C: low-density lipoprotein cholesterol; HDL-C: high-density lipoprotein cholesterol; TG: triglycerides; AST: aspartate aminotransferase; ALT: Alanine aminotransferase; γ-gt: gamma-glutamyl transferase; CPK: creatine phosphokinase. Data presented as mean ± SEM.

No change was observed in BMI during treatment ($t_0$ vs. $t_8$, 25.2 ± 0.6 vs. 25.3 ± 0.5 Kg/m$^2$, *p* = 0.12). A significant decline was observed in both TC ($t_0$ vs. $t_8$, 258.9 ± 4.0 mg/dL vs. 212.7 ± 4.5 mg/dL, *p* < 0.0001) and LDL-C concentrations ($t_0$ vs. $t_8$, 173.8 ± 3.5 mg/dL vs. 129.0 ± 4.5 mg/dL, *p* < 0.0001) (see Figures 1 and 2). The LDL-C mean decline was −42.4 ± 3.6 (95% confidence interval: −49.7 to −35.0, r$^2$ = 0.8051). Overall, there was a mean reduction of 25.6% in LDL-C levels (range −4 to −58%); 20 out of 37 patients (54.1%) had a decline in serum LDL-C ≥ 25% and 13/37 (35.1%) exhibited a decline ≥ 30%.

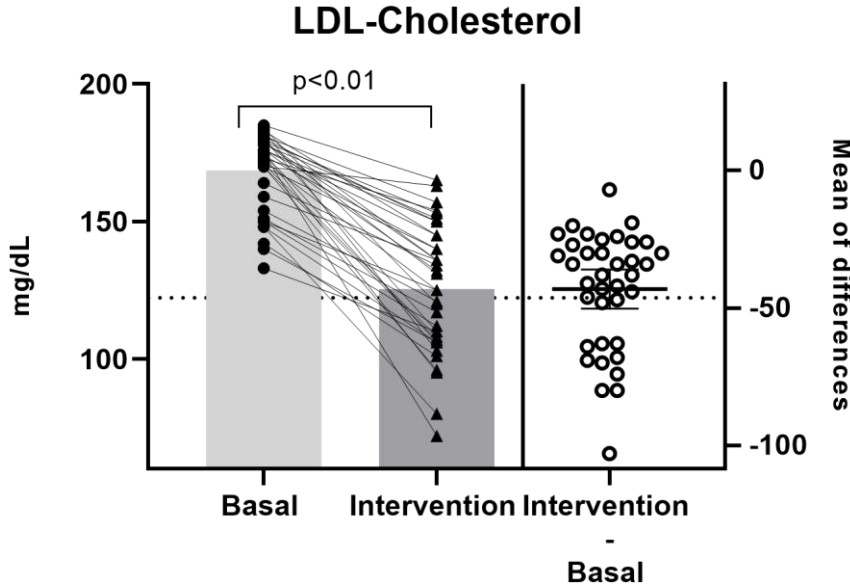

**Figure 1.** Low-density lipoprotein-cholesterol concentration changes (mg/dL). Baseline (black circles) and after 8 weeks of intervention (black triangles). Scatter plot of means of differences (white circles) with 95% confidence interval (CI). $p < 0.01$, paired $t$-test.

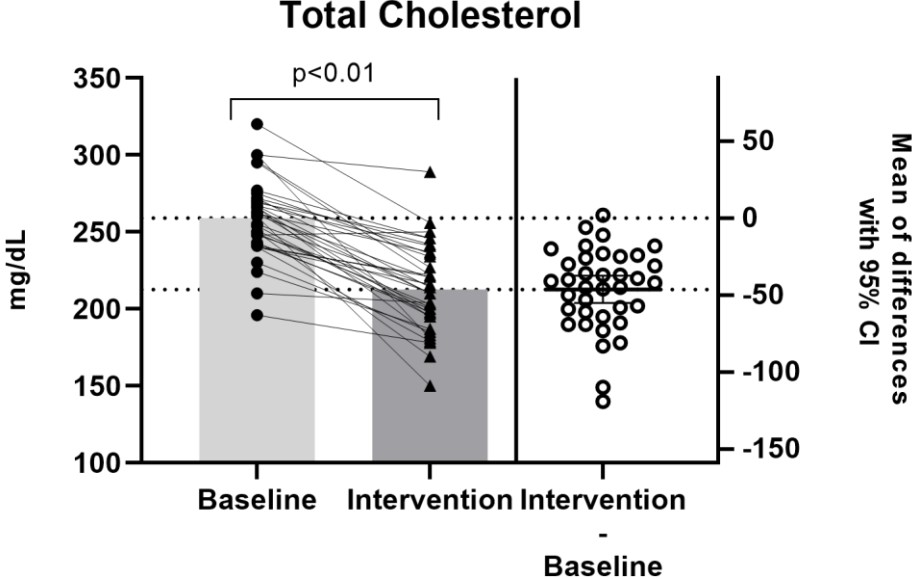

**Figure 2.** Total-cholesterol concentrations at baseline and after 8 weeks of intervention (mg/dL). Baseline (black circles); intervention (black triangles). Scatter plot of means of differences (white circles) with 95% confidence interval (CI). $p < 0.01$, paired $t$-test.

No significant difference was observed in HDL-C levels ($t_0$ vs. $t_8$, $63.3 \pm 2.5$ mg/dL vs. $63.1 \pm 2.6$ mg/dL, $p = 0.90$). A small but statistically significant decrease was detected in TG levels ($t_0$ vs. $t_8$, $127.0 \pm 12.2$ mg/dL vs. $117.0 \pm 9.2$ mg/dL, $p = 0.012$) (Table 3). No increase in liver function test or muscle enzyme values was detected before and after treatment in our cohort: AST ($27.43 \pm 0.9$ vs. $29.17 \pm 1.1$, U/L, $p = 0.24$), ALT ($30.43 \pm 0.9$ vs. $39.13 \pm 1.2$, U/L, $p = 0.40$), $\gamma$-gt ($29.18 \pm 1.1$ vs. $29.43 \pm 0.9$, U/L, $p = 0.71$), and CPK ($75.9 \pm 3.2$ vs. $75.1 \pm 3.3$, IU/L, $p = 0.70$).

**Table 3.** Comparison of variables before and after 8 weeks of intervention.

|  | **Basal** | **Treated** |
|---|---|---|
| BMI † | 25.3 ± 0.54 | 25.2 ± 0.57 |
| TC † ** | 258.9 ± 4 | 212.7 ± 4.5 |
| HDL-C † | 63.1 ± 2.6 | 63.3 ± 2.5 |
| LDL-C † ** |  |  |
| Absolute values | 173.8 ± 3.45 | 129 ± 4.53 |
| Difference |  | −42.35 ± 4.15 |
| TG ‡ * | 127 ± 12.19 | 117 ± 9.22 |

TC, total cholesterol; HDL-C, HDL-cholesterol; LDL-C, LDL-cholesterol; TG, triglycerides; %; † paired t test; ‡ Wilcoxon test; * $p < 0.05$ and ** $p < 0.01$. Data presented as mean ± SEM.

## 4. Discussion

The recent guidelines pertaining to cardiovascular risk reduction issued by all relevant professional organizations mandate treatment with lipid-lowering medications in patients at increased risk for cardiovascular disease, based on a personalized calculation of that risk (such as the ASCVD Plus risk estimator) [22]. The initial strategy, which is common in all these guidelines, consists of treatment with hydroxymethylglutaryl-coenzyme A (HMG-CoA) reductase inhibitors, i.e., statins, in patients at risk of cardiovascular events deemed adequate for the above treatment, independent of their baseline cholesterol concentration. The 2019 AHA/ACC guidelines, which are endorsed by a multitude of other professional organizations worldwide, have set the treatment threshold of the 10-year risk for cardiovascular events at 7.5%. From that risk upwards, treatment with statins is required and, if needed, additional medications aiming for a reduction in LDL-C, which is the most frequently used predictor of cardiovascular risk. Nevertheless, despite these guidelines, a significant number of patients present with elevated concentrations of LDL-C, yet they do not reach that threshold in order to receive treatment. Additionally, a large number of patients worldwide present with cardiovascular events despite not qualifying for primary prevention prior to those events based on these estimations [21,26,27]. The risk estimation models have always sought to equilibrate the risks associated with a treatment modality and the benefits in the form of risk reduction obtained with that treatment.

Since all medications have significant risks, in addition to their proven benefits, their use is clinically justified only in patients for whom the benefits outweigh the risks. In patients with elevated cholesterol concentrations who do not qualify for treatment based on this equation, what should our strategy be in order to achieve clinically significant cardiovascular disease risk reduction? The answer is, in our opinion, that strategies should be employed which could have a beneficial effect on lipid concentrations and, therefore, could have cardiovascular risk reduction side-benefits when the risks attributed to such interventions are small. In this regard, multiple studies have looked into the possibility of correcting dyslipidemias with the use of dietary supplements, particularly in those individuals with low-to-moderate risk (<7.5%), as an alternative to statins, either because some patients may have milder forms of hypercholesterolemia or in patients who may have experienced side effects from the current standard regimens. However, the safety profile of such supplements has not, to date, been investigated adequately.

### 4.1. RYR-Based Supplements Efficacy

The extracts of RYR are currently the most effective cholesterol-lowering nutraceuticals [28]. RYR is produced via the process of fermenting rice with yeasts (mainly Monascus purpureus); it has a long tradition in China where it is employed in cooking as food coloring, while it is also used medicinally to assist digestion and blood circulation. Daily consumption results in a reduction in LDL-C plasma levels by up to 15 to 25% within 6 to 8 weeks [29,30]. The decrease in LDL-C is accompanied by a proportional decrease in TC

and non-HDL-C, plasma apolipoprotein B, and high-sensitivity C-reactive protein [21]. Some trials indicate that RYR use is associated with improvements in endothelial function and arterial stiffness, while a long-term study provides some evidence of its role in the prevention of cardiovascular events [30,31]. Moreover, as recently suggested by the International Lipid Expert Panel, consuming combinations of nutraceuticals with different lipid-lowering actions, especially in conjunction with a healthy lifestyle, could be a valid alternative to statins for preventing coronary heart disease (CHD) in patients with moderate hypercholesterolemia, and in some patients with statin intolerance [31,32]. In particular, the interaction between RYR and natural products with different mechanisms of action may have a synergistic effect. For example, RYR's potential to inhibit the HMG-CoA reductase enzyme might be neutralized by coupling it with other nutraceuticals, such as plant sterols, to increase lipid excretion in the bowel [30].

The lipid-lowering effects of RYR have been determined via a number of meta-analyses of randomized controlled clinical trials (RCTs). The most recent, by Gerard et al., comprised 20 studies with RYR doses ranging between 1200 and 4800 mg/day and containing 4.8–24 mg of monacolin K. The meta-analysis showed that RYR lowered LDL-C by an average of 39.4 mg/dL following 2–24 months of treatment by comparison with a placebo. The latter LDL-C lowering impact was not different from that of low/moderate-intensity statins, including 40 mg pravastatin and 20 mg lovastatin [30].

Nevertheless, the dose of monacolin varied in the enrolled studies, while cumulative results cannot be comparable with ours since different lipid-lowering substances were included in the various forms of neutraceuticals used. In our cohort, the use of a neutraceutical product based on RYR produced a significant reduction in TC and LDL-C (by approximately 25%) within 6 weeks of therapy. Of interest, in a study with a similar design and the same dose of monacolin (10 mg) for a short period (1 month), Mazza et al. [33] reported a significant reduction in the levels of TC and LDL-C (9.2% and 12.3%, respectively), compared to a placebo in patients treated with a neutraceutical containing policosanols, resveratrol, and chromium picolinate. Regarding TG levels, we observed a slight but significant reduction with treatment (12.7%), while in the abovementioned study, there was no significant reduction (−24.5% in the treated group vs. −20.8% in the placebo group). Once again, differences in the lipid-lowering effect of the two supplements may be attributed to the different composition and the absence of a control group in our study. In line with our reports, they failed to detect any significant difference in the levels of HLD-C with treatment [33].

### 4.2. Safety of RYR-Based Supplements

In approximately 5–10% of adults, the use of statins is associated with a range of muscle side effects, including the commonly described muscle aches (myalgia), muscle cramps, and, more rarely, muscle weakness, while on rarer occasions, rhabdomyolysis has been described as well [34]. Despite the statin-like mechanism of action, the statin-associated muscle symptoms (SAMS) risk related to monacolin K is minimal when doses equal to 3–10 mg daily have been used, with the exception of mild myalgia in previously severely statin-intolerant patients [30].

However, several previous studies have reported that the profile of adverse effects with RYR was similar to that of lovastatin; through consultation of four sources of case reports (the World Health Organization (WHO), the French Agency for Food, Environmental and Occupational Health & Safety (ANSES), the Italian Surveillance System, and the Food and Drug Administration (FDA)), the most important targets for adverse events were as follows: musculoskeletal and connective tissue (29.9–37.2% of cases, including 1–5% of rhabdomyolysis); liver (9–32%); nervous system (when reported, 12.8–26.9%); gastrointestinal tract (12–23.1%); and skin and subcutaneous tissue (8–17.3%) [35–38].

Since it has been suggested that a decrease in mitochondrial CoQ10, specifically in ubiquinone, may be a mechanism of SAMS, a number of formulations of CoQ10 supplements are on the market to prevent and treat SAMS. On the other hand, current study

results are at present ambiguous, with some data suggesting that CoQ10 supplementation may lower SAMS incidence and severity [39] and others observing no health benefit with supplementation [40].

Interestingly, in our cohort, no subject complained of SAMS, although five patients had previously discontinued therapy with low-intensity statins (simvastatin 10 mg) due to myalgias. This promising outcome could be attributed to the combination of a protective effect from Q10 supplementation resulting in the minimal adverse effect of monacolin on muscles.

Effects on the liver are among the most frequent and severe adverse effects reported in association with the intake of RYR; Mazzanti et al. [41] described 10 cases of hepatobiliary disorders associated with the intake of RYR. Two of them were considered unassessable/unlikely. In addition, two more studies [42,43] reported a total of four cases presenting with increased transaminases after the use of food supplements containing RYR for a few months. In our study, there was no significant elevation in liver enzymes in any subject during the 8 weeks of treatment. A systematic literature review and meta-analysis of randomized clinical trials with RYR-based products summarized reported adverse effects [21]: gastrointestinal disorders (diarrhea, GI discomfort, and other symptoms), musculoskeletal (arthralgia and weakness), laboratory value alterations (leukocytosis, leukopenia, and hyperglycaemia), infectious problems (influenza, urinary tract, and pneumonia), immunologic problems (rash, alopecia, and allergic reactions), general problems (dizziness, malaise, and fatigue), central nervous system disorders (headache), and cardiovascular disorders (QT prolongation, uncontrolled hypertension, oedema, and erectile dysfunction). Of note, one of our patients experienced palpitations and therapy was discontinued, although further evaluation did not reveal any cardiac problems.

### 4.3. Efficacy of GSE, Polyphenols, and Vitamin B Supplements

There are as yet no conclusive data on the effect of GSE on lipid profile. In one study, while GSE supplementation appeared to considerably decrease LDL-C serum levels and TG concentrations, it had no impact on TC and HDL-C concentrations [44]. Meanwhile, though the same study/other studies observed no significant effect of GSE supplementation on circulating TC and HDL-C levels, there have been reports of significant decreases in these lipids in studies including <10 weeks of intervention and also in those administering dosages of <300 mg/d of GSE [44].

In our study, a reduction in LDL-C was observed with the treatment that was comparable to that seen in studies employing RYR alone, rendering GSE as a likely neutral agent in our supplement.

Dietary polyphenols have been able to positively modify a number of cardiovascular risk markers, such as blood pressure (BP), endothelial function, and plasma lipids in prior studies [45,46]. A randomized, double-blind trial (without a control group) described a number of positive effects of a phenolic-rich OLE on BP and several associated vascular and metabolic measures (reductions in plasma TC, LDL-C, and TG) [47]. There is a paucity of data regarding the potential effects of vitamin B complex on lipid metabolism. Two studies reported some beneficial effects in a healthy elderly population; specifically, B vitamins increased HDL-C by 3.4% after 6 months (0.04; 95% CI $-0.02$, 0.10; $p = 0.155$) and by 9.2% after 12 months (0.11; 95% CI 0.04, 0.18; $p = 0.003$) [48]. A randomized, double-blind, placebo-controlled trial including adults with stable coronary artery disease produced a significant improvement in the concentrations of serum TG, TC, and HDL-C ($p < 0.05$) after 12 weeks of intervention of supplementation with B vitamins [49]. To what extent this beneficial effect might have impacted the observed improvements in lipid profile in our cohort is unknown.

The European Cardiology Society and European Atherosclerosis Society guidelines have since 2011 suggested the use of RYR extract in the management of mild hypercholesterolemia [50] and, recently, the International Lipid Expert Panel also concurred with the latter recommendation [29]. However, some concerns have lately been raised as to the

safety of RYR in frail patients [51]. Meanwhile, there are still concerns with the generalized use of multi-ingredient botanical preparations, similar to that used in our study. In our view, the main problems involved are the lack of data on the bioactivity of components of RYR, other than monacolin K, and the effects of concomitant consumption of RYR-based supplements with foods or drugs inhibiting cytochrome P450 3A4 (CYP3A4), which could have significant interactions with other commonly used medications.

*4.4. Limitations*

Our study has several limitations. The trial period, though rather short, was adequate to note significant LDL-C, TC, and serum TG lowering in our study participants. On the other hand, the effect on lipid biomarkers of supplements that need a longer duration of use may not have been captured due to the relatively short study period. Moreover, the absence of a control group also weakens the interpretation of our results given that the effect of dieting on lipid profile may be miscalculated. Likewise, a trial of this size and scope is not capable of determining the impact of supplements on cardiovascular outcomes, even though we were able to measure biomarkers that are known to predict and affect cardiovascular health.

## 5. Conclusions

Our results confirm the clinically relevant lipid-lowering properties of monacolin K when used in conjunction with CoQ10 and potent antioxidants in the form of polyphenols. Furthermore, we found that this combination had an excellent safety profile in terms of liver toxicity and muscle side effects. Taking all the above into consideration, we therefore suggest that RYR combined with CoQ10 and polyphenols could represent an invaluable therapeutic tool to support lifestyle modifications in managing dyslipidemia in low-risk patients, while it could also serve as an alternative or add-on in those patients who cannot reach their LDL-C goals with the highest-tolerated dose of statins and/or other LDL-C-lowering medications.

**Author Contributions:** N.A., A.B. and S.L. designed the research; N.A., R.D.P., I.A., P.A. and S.L. analyzed the data; N.A. wrote the paper; N.A., R.D.P., V.T. and S.L. had primary responsibility for the final content. All authors have read and agreed to the published version of the manuscript.

**Funding:** This research received no external funding.

**Institutional Review Board Statement:** This study was multicenter-approved by the Institutional Review Board Committee at Athens Medical Centre, Athens, Greece.

**Informed Consent Statement:** All subjects gave their informed consent for inclusion before they participated in the study.

**Data Availability Statement:** The datasets used and analyzed during the current study are available from the corresponding author upon reasonable request.

**Conflicts of Interest:** The authors declare no conflict of interest.

## Abbreviations

| | |
|---|---|
| ALT | Alanine aminotransferase |
| ANSES | French Agency for Food, Environmental and Occupational Health & Safety |
| AST | aspartate aminotransferase |
| ASCVD | Atherosclerotic cardiovascular disease |
| BMI | Body mass index |
| BP | Blood pressure |
| Co Q10 | Coenzyme Q10 |
| CHD | Coronary heart disease |
| CK | creatine kinase |
| CYP3A4 | Cytochrome P450 3A4 |
| $\gamma$-gt | gamma-glutamyl transferase FDA— Food and Drug Administration |

|       |                                                                  |
|-------|------------------------------------------------------------------|
| GSE   | grape seed extract                                               |
| HDL-C | High-density lipoprotein cholesterol                             |
| HgbA1c| Hemoglobin A1c                                                   |
| HMG-CoA | hydroxymethylglutaryl-coenzyme A                               |
| LDL-C | Low-density lipoprotein cholesterol                              |
| MDRD GFR | Modification of Diet in Renal Disease calculated glomerular filtration rate |
| OLE   | Olive leaf extract                                               |
| RYR   | Red yeast rice extract                                           |
| SAMS  | Statin-associated muscle symptoms                               |
| TC    | Total cholesterol                                               |
| TG    | Triglycerides                                                   |
| WHO   | World Health Organization                                       |

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
