# Peer review of "Efficacy and Safety of Monacolin K Combined with Coenzyme Q10, Grape Seed, and Olive Leaf Extracts in Improving Lipid Profile of Patients with Mild-to-Moderate Hypercholesterolemia: A Self-Control Study"

_nutraceuticals, doi:10.3390/nutraceuticals3010001_

Round 1

Reviewer 1 Report

In this manuscript, the authors describe and assess the lipid-lowering activity and safety of the greek nutraceutical Arichol® on 37 (prior 43) participants (mainly women). The study's rationale is evident with the following results being well discussed by the authors. 

The manuscript is well-written and fits with the journal's expectations. Therefore, it is suitable for publication as it is.

However, I do have some requests for the authors for future studies involving this nutraceutical. To validate these results (as for now, they are not), the authors will need another investigation with a different protocol involving participants receiving no medications and placebos for claiming benefits of Arichol® (as studies following such protocols are cited in the manuscript's discussion). 

Author Response

Response: Thank you for your comments. We have recently registered a new protocol involving both a placebo group and a group with a lower dose of monacolin 3mg/day. This prospective study will be completed on February 2021.

Reviewer 2 Report

The authors of the manuscript entitled "Efficacy and safety of Monacolin K combined with Coenzyme 2 Q10, Grape seed and Olive leaf extracts in lipid profile of patients with mild to moderate hypercholesterolemia: a self-control study" assess the lipid-lowering efficacy and safety of a novel dietary supplement containing monacolin K, combined with Coenzyme Q10, Grape seed and Olive tree leaf extracts (Arichol®) on the lipid profile of healthy volunteers. In my opinion, the study design and concept are good. The aim of the study was clearly defined. However, I have comments and suggestions regarding to the methods and results of the study.

 Abstract

Line 21 - I refuse to use the term "healthy" volunteers. In my opinion, the absence of diagnosis in increased LDL is not a guarantee of health. I recommend to consider using the "healthy" label and look at the definition of health at least according to the WHO.

Rows 29-30 - I recommend to include specific values ​​of changes instead of percentage values.

 Introduction

In my opinion, this section is disproportionately short with limited content focused on the issue. There are no important knowledge and associations related to the issue. The theoretical basis and overview is very brief.

 Line 50 - healthy volunteers - my opinion is the same as above.

 Materials and Methods

Line 56 - The sentence confirms my opinion and at the same time refutes the claims that the participants were healthy volunteers.

Rows 78-88 - In this section, you state that volunteers were instructed on lifestyle adjustment. In my opinion, this was a fundamental intervention in the lifestyle of volunteers, as long as their eating habits and physical activity were different than they were recommended. Where are you sure that volunteers followed your recommendations and followed and practiced everything? How did you verify compliance with these regulations? Did you use a structured questionnaire? Or FFQ? Or a questionnaire about physical activity? About the intensity of physical activity? What medicines or other dietary supplements were used/could be taken by participants that could theoretically affect the results?

 Line 92 - Why was a highly carbohydrate diet recommended to participants? What method did you realize a measurement of body weight and height? What was used for biochemical analysis - blood serum or plasma?

 Resuls

Line 124 - What was the cause of the loss of another 5 subjects? Was it related to intervention?

 I regret that this section is very poor and insufficient with fundamental mistakes. First of all, the basic table with the description of the research group is lacking.

Line 129 - mean age was 55.9 years (range 32-72 years). How did you determine the 10-year risk in people under 40 years of age? There is no mention of this in the manuscript.

Figures and charts are, in my opinion, too simple without the elements of science. If possible, use another graphic format, if possible for both graphs.

Line 130 - Significant decline? OK, but in the text you also mention something completely different that does not match the data in Table 2.

 I recommend this section to completely re-stylize and correct fundamental errors and supplement the minimum basic characteristics of the research group.

 Therefore, I recommend giving the authors a chance to revise the manuscript after major revision.

Author Response

Reviewer 2

  1. Abstract

  1. - Line 21 –

Response: We agree with your comment and phrase “healthy” was omitted and replaced: “on the lipid profile of adults with moderate cholesterol elevations and absence of concomitant risk factors”

  1. Rows 29-30 - I recommend to include specific values ​​of changes instead of percentage values.

Response: We agree, values were changed

  1. Introduction

  1. In my opinion, this section is disproportionately short with limited content focused on the issue. There are no important knowledge and associations related to the issue. The theoretical basis and overview is very brief.

Response: This section was rewritten according your suggestions

  1. Line 50 - healthy volunteers - my opinion is the same as above.

Response: Was corrected

  1.  Materials and Methods

  1. Line 56 - The sentence confirms my opinion and at the same time refutes the claims that the participants were healthy volunteers.

Response: We agree and clarified this issue as mentioned above

  1. Rows 78-88 - In this section, you state that volunteers were instructed on lifestyle adjustment. In my opinion, this was a fundamental intervention in the lifestyle of volunteers, as long as their eating habits and physical activity were different than they were recommended. Where are you sure that volunteers followed your recommendations and followed and practiced everything? How did you verify compliance with these regulations? Did you use a structured questionnaire? Or FFQ? Or a questionnaire about physical activity? About the intensity of physical activity? What medicines or other dietary supplements were used/could be taken by participants that could theoretically affect the results?

Response: unfortunately there was no control group in order to validate the role of life-style intervention in our cohort. However, as mentioned in this section, compliance on diet recommendations was evaluated by structured questioner on final visit. No other medications (nor supplements) that could affect lipids were used during the study period.

  1. Line 92 - Why was a highly carbohydrate diet recommended to participants?

Response:   We advised participant to follow a short time high carbohydrates diet in order to avoid lipids consumption before the blood exams ( approximately 300gr in the total consumption of 2000 calories).  Short term glycemic load however, is inversely associated with the blood levels of total cholesterol, LDL-C and HDL-C and moderately increases the overall TC:HDL-C ratio. Therefore, we added the relevant reference in the manuscript

  1. What was used for biochemical analysis - blood serum or plasma?

Response: Serum was separated by centrifugation (3000 rmp for 15 min at 4 °C) and stored at -80 °C refrigerator for further analysis. It is now clarified in the manuscript

  1. Results

  1. Line 124 - What was the cause of the loss of another 5 subjects? Was it related to intervention?

Response: 4 persons refused to go back at the end of the trial and 1 person was disengaged because of relocation 

  1.  I regret that this section is very poor and insufficient with fundamental mistakes. First of all, the basic table with the description of the research group is lacking.

Response: Table 2 now illustrates the descriptive characteristics of the study population

  1. Line 129 - mean age was 55.9 years (range 32-72 years). How did you determine the 10-year risk in people under 40 years of age? There is no mention of this in the manuscript.

Response: Thank you for the comment it is very important and we clarified that in the section of “materials and methods”. The 10 year risk was determined only in patients over 40 years old. In subjects under 40 years old , only ldl levels were taken into account. The number of patients younger than 40 years old is mentioned in the results.

  1. Figures and charts are, in my opinion, too simple without the elements of science. If possible, use another graphic format, if possible for both graphs.

Response: Both figures were changed and we hope that the new format represents finding in a better manner. 

  1. Line 130 - Significant decline? OK, but in the text you also mention something completely different that does not match the data in Table 2.

Response: Thank you for the comment. There was a typographical error and data were written inversely in the parenthesis. It is now corrected

Reviewer 3 Report

Introduction

Please include in the introduction some information about relevant issues on the use of grape seeds, olive leaves, and vitamin B to enforce the study of these in the main objective

 In table I, it is not necessary to write in bold font the rice description (line 89), all the treatment suggest will be in the same font. Title is ok to be bolded.

On page 3, lines 96-97, be more specific, how long it takes to perform the analysis, because “until further analysis “ is not clear, and can jeopardize all your results, because blood samples to be trustable need to be analyzed no later than 4 hour of extraction, if this was not case please explain better this procedure. Perhaps only maintained in refrigeration conditions until arrival to the laboratory facilities for separation of serum, plasm, etc.

Also, the term “simultaneously” it is not clear.

On the same page 3, On line 137, remove mean±SEM.

On lines 137-138, once again the values are inverted, an remove “mean±SEM”

 On line 139, I suggest to write “before and after our cohort”, and eliminate all the before and after words in parenthesis. Please check that the values are well written in the parenthesis.

On page 6, table 2, I will recommend to redo it, using the biomarkers names as files and the mean and SEM as columns. It will be easier to follow the results shown in the table. Please give a more descriptive Title also to the table.

All discussion lacks of contrast the relevant results of the study in comparison to other previously reported, It is more like am introduction.

I will recommend rewriting this part and use also the important or studies that can be compared with your results.

Author Response

  1. Introduction
  2. Please include in the introduction some information about relevant issues on the use of grape seeds, olive leaves, and vitamin B to enforce the study of these in the main objective

Response: the section was rewritten and more relevant studies were added

  1. in table I, it is not necessary to write in bold font the rice description (line 89), all the treatment suggest will be in the same font. Title is ok to be bolded.

Response: Corrected

  1. On page 3, lines 96-97, be more specific, how long it takes to perform the analysis, because “until further analysis “ is not clear, and can jeopardize all your results, because blood samples to be trustable need to be analyzed no later than 4 hour of extraction, if this was not case please explain better this procedure. Perhaps only maintained in refrigeration conditions until arrival to the laboratory facilities for separation of serum, plasm, etc.Also, the term “simultaneously” it is not clear.

Response: It is now clarified in the text

  1. On the same page 3, On line 137, remove mean±SEM.

Response: It has been removed

  1. On lines 137-138, once again the values are inverted, an remove “mean±SEM”

Response: It has been removed

  1. On line 139, I suggest to write “before and after our cohort”, and eliminate all the before and after words in parenthesis. Please check that the values are well written in the parenthesis.

Response: It has been removed

  1. On page 6, table 2, I will recommend to redo it, using the biomarkers names as files and the mean and SEM as columns. It will be easier to follow the results shown in the table. Please give a more descriptive Title also to the table.

Response:Table was reformed according to your suggestion   

  1. All discussion lacks of contrast the relevant results of the study in comparison to other previously reported, It is more like am introduction. I will recommend rewriting this part and use also the important or studies that can be compared with your results.

Response : This section was rewritten with an effort to refer relevant studies

Round 2

Reviewer 2 Report

Dear authors, thank you for answering my questions and making the edits.

Author Response

 Thank you very much for your comments and your contribioution  to improve our manuscript 

Reviewer 3 Report

Page 3, on line 124-125. Please review that the equipment be correct, in my knowledge this equipment do not run colorimetric analysis, as you establish for lipids. The same for lines 126 and 127.

Page4,  on table 2, in the title, I suggest to use singular for population, because it is one population composed by female and males, but only that.

In the main text of the table please review the correct form to abbreviate the units, dl It must be dL, Kgr must be corrected as kg, and others details, please check the attached document.

Remove the range from the variable column, if you think it is relevant integrate this in to the text.

Check also the correct form to abbreviate the gamma……..enzyme.

In line 173, please add a space between the 4.0mg

On Figure 1 and 2, correct the units for dl, it must be dL

On line 193, Table3, why is so relevant to include the difference found for LDL as % and difference and not for TC, both markers were significant different. I recommend leaving only the crude data for each variable and in the text explain this difference.

Author Response

  1. Page 3, on line 124-125. Please review that the equipment be correct, in my knowledge this equipment do not run colorimetric analysis, as you establish for lipids. The same for lines 126 and 127.

Response: you are absolutely right; the immune Siemens does not calculate lipids profile.   It was clarified by our laboratory and corrected in the text. 

  1. Page4,  on table 2, in the title, I suggest to use singular for population, because it is one population composed by female and males, but only that.

Response: It was a typo mistake and we corrected it

  1. In the main text of the table please review the correct form to abbreviate the units, dl It must be dL, Kgr must be corrected as kg, and others details, please check the attached document.

Response: All abbreviations were corrected.  

4, Remove the range from the variable column, if you think it is relevant integrate this in to the text.

Response: It is removed  

  1. Check also the correct form to abbreviate the gamma……..enzyme.

Response: All abbreviations were corrected.  

  1. In line 173, please add a space between the 4.0mg

Response: corrected

  1. On Figure 1 and 2, correct the units for dl, it must be dL

Response: corrected

  1. On line 193, Table3, why is so relevant to include the difference found for LDL as % and difference and not for TC, both markers were significant different. I recommend leaving only the crude data for each variable and in the text explain this difference.

Response: % difference for LDL is removed from the table and mentioned only in the text

Thank you very much for your comments and your contribution  to improve our manuscript